# Impact of Chitosan on the Mechanical Stability of Soils

**DOI:** 10.3390/molecules27072273

**Published:** 2022-03-31

**Authors:** Agnieszka Adamczuk, Grzegorz Jozefaciuk

**Affiliations:** Department of Physical Chemistry of Porous Materials, Institute of Agrophysics of Polish Academy of Sciences, Doswiadczalna 4 Str., 20-290 Lublin, Poland; jozefaci@ipan.lublin.pl

**Keywords:** aggregate, chitin derivatives, food wastes, stress, strain, Young’s modulus

## Abstract

Chitosan is becoming increasingly applied in agriculture, mostly as a powder, however little is known about its effect on soil mechanical properties. Uniaxial compression test was performed for cylindrical soil aggregates prepared from four soils of various properties (very acidic Podzol, acidic Arenosol, neutral Fluvisol and alkaline Umbrisol) containing different proportions of two kinds of chitosan (CS1 of higher molecular mass and lower deacetylation degree, and CS2 of lower molecular mass and higher deacetylation degree), pretreated with 1 and 10 wetting–drying cycles. In most cases increasing chitosan rates successively decreased the mechanical stability of soils that was accompanied by a tendential increase in soil porosity. In one case (Fluvisol treated with CS2) the porosity decreased and mechanical stability increased with increasing chitosan dose. The behavior of acidic soils (Podzol and Arenosol) treated with CS2, differed from the other soils: after an initial decrease, the strength of aggregates increased with increasing chitosan amendment, despite the porosity consequently decreasing. After 10 wetting–drying cycles, the strength of the aggregates of acidic soils appeared to increase while it decreased for neutral and alkaline soils. Possible mechanisms of soil–chitosan interactions affecting mechanical strength are discussed and linked with soil water stability and wettability.

## 1. Introduction

Due to its non-toxicity, bacteriostatic properties, and polycationic nature, chitosan has found many industrial, medical, pharmaceutical, and environmental applications [1,2,3]. The availability of industrial quantities of chitosan in the late 1980s enabled it to be tested in agriculture [4]. Up to date, chitosan has been increasingly applied for the improvement of the degradative capacity of contaminated soils by introducing specific consortia of microorganisms (bioaugmentation), providing nutrients to the chemical degrading indigenous microorganisms present at the contaminated sites (biostimulation), biochemical degradation of xenobiotic compounds. Chitosan is used as a carrier for microbial communities that can be encapsulated for further use and for providing resistance to the enzymes from harsh environmental conditions like pH and temperature. It is applied as an antiviral, antibacterial, antifungal and antinematode agent, for seeds coatings, and plant protection by controlling the spreading of diseases [5,6]. Chitosan has been proven to stimulate plant growth and yield, and to induce abiotic and biotic stresses tolerance in various commodities [7,8]. Besides that, chitosan has been employed in soil as a plant nutrient and has shown great efficacy in combination with other industrial fertilizers without affecting the soil’s beneficial microbes. Furthermore, it is helpful in reducing fertilizer losses thus reducing the overuse of synthetic fertilizers during crop production, which is important in keeping environmental pollution under check. More recently chitosan is applied as a valuable delivery system for fertilizers, herbicides, pesticides, and micronutrients for crop growth promotion by maintaining balanced and sustained nutrition [1,4,9,10]. Various composite or copolymer systems containing chitosan are used to improve the soil’s water-holding capacity [11,12]. Chitosan is also applied for the removal of various types of contaminants from soils [13,14].

Chitosan introduced into a soil in one or more above ways can, as any other biopolymer, interact with soil components in different ways, including adsorption of polymer molecules on surfaces of soil components, covering soil particles with a thin polymer film, formation of polymer ties connecting neighboring particles, adhesion, hydrogen bonding or bridging of soil particles via polyvalent counterions [15,16]. Direct binding of a charged polymer to negatively charged soil components (clay minerals, silica, feldspars, organic matter) with electrostatic forces seems to be a rare feature of chitosan, because, contrary to the majority of natural biopolymers applied for soil treatment (tragacanth, arabic, karaya, gellan, carrageenan, locust bean, xanthan, guar and/or tamarind gums, agar, pectin, alginate, arabinans, amylase, lipids, kasein, cellulose) having anionic or non-ionic character [17,18,19], it carries a positive charge in a broad range of neutral and acidic pH values [20]. So strong interactions of chitosan with soil components imply that the addition of chitosan should increase soil mechanical resistance leading to soil stabilization, consolidation and hardening. However, among various biopolymers implemented in recent years for soil stabilization, erosion mitigation, and dust control, chitosan has been applied rarely. Huang et al. [17], in their comprehensive review of the application of various polymers for soil stabilization, mentioned only a single application of chitosan. Most probably the application of chitosan for soil stabilization is limited by its very poor solubility in water which changes to some extent depending on molecular weight, deacetylation degree and crystal structure [21,22,23] and/or by its high biodegradation rate (studies of Mostafa et al. [24] showed the fastest biodegradation of chitosan as compared to the other studied biopolymers). Apart from the successful application of various interpolymer complexes, nanoparticles or composites including chitosan [25,26,27,28,29,30,31], a positive effect on soil stabilization by pure chitosan has been achieved only for its highly dispersed jellified forms (chitosan dissolved usually in organic acids). Orts et al. [28] applied chitosan gel for hardening eroded furrows. They observed a strong effect of soil stabilization in laboratory conditions, however, under natural conditions, the effect was observed only in the initial part of the furrow, which was explained by the very strong binding of chitosan to soil components in the area close to the zone of its application. Shariatmadari et al. [32] studied the effects of chitosan dissolved in acetic acid on sandy soil stabilization by using unconfined compression tests. The unconfined compression tests conducted after 7, 14 and 28 days showed the strength of samples treated in dry conditions was higher than for samples cured in saturated water conditions. In contrast, Hataf et al. [27] who studied uniaxial compression of sand–chitosan gel mixtures, observed that chitosan improved soil strength, but its effect decreased with the reduction of water. They concluded that the additional strength that could be achieved with the addition of chitosan to the soil fades as the soil becomes dry. Aguilar et al. [33] investigated the feasibility of using chitosan dissolved in acetic acid as an admixture, or as an external coating, for earthen constructions to improve their resistance in terms of both their key mechanical properties and, more specifically, during the occurrence of water-induced degradation. They showed that in both instances the use of low concentrations of chitosan can significantly improve performance.

However, for agricultural purposes, the application of chitosan gels containing huge amounts of acid may be dangerous for the soil environment because of the potential for inducing aluminum toxicity, mineral destruction, nutrient leaching or heavy metal mobilization [34,35,36]. In agricultural applications, chitosan is usually introduced into the soil mostly as powders (pure components and admixtures), and also as membranes (chitosan coatings) or particulates (suspensions) [1,2,10,37,38,39,40,41]. Despite this, the literature brings extremely small information on the mechanical properties of soils amended with chitosan, not in a gel but in a powder form. After an extensive search, the authors found only a single paper on this topic by Soldo et al. [42], who studied the mechanical properties of Piedmont well-graded sand with silt, amended with various biopolymers, including chitosan. They mentioned that chitosan was applied in a form as it was supplied by the manufacturer (probably as the powder). After five days of curing, chitosan did not lead to any further increase in the compressive strength even though the specimens with chitosan had the highest compressive strength right after the mixing. Therefore they excluded chitosan from further research of this type.

To fill this knowledge gap, this paper studies the mechanical stability of a few soils amended with different doses of chitosan powder by a uniaxial compression test. Because various chitosan–soil binding mechanisms contribute differently depending on polymer solubilization, concentration, charge, conformation, molecular weight and molecule size, that in turn depend on the ambient solution pH and ionic strength [43,44,45], four soils of different pH values and two types of chitosan, differing in molecular masses and deacetylation degrees were selected for the testing. Since cyclic wetting–drying affects biopolymer treated soils [46,47,48] and is used as a measure of the durability of biopolymers [46], the studied soils were subjected also to one and ten wetting–drying cycles.

## 2. Results

The addition of chitosan of high molecular mass and low deacetylation degree (CS1, see M&M section) in the rates of 0.5, 1, 2, 4 and 8% decreased, in general, the mechanical strength of four studied soils: very acidic Podzol, acidic Arenosol, neutral Fluvisol and alkaline Umbrisol. The intensity of this decrease depended on chitosan dose and the number of wetting–drying cycles with which the soil–chitosan mixtures were pretreated, as is seen from exemplary stress–strain curves of soil cylindrical aggregates shown in Figure 1.

The addition of chitosan of lower molecular mass and higher deacetylation degree (CS2, see M&M section) in the same rates of 0.5, 1, 2, 4 and 8% decreased the mechanical strength of Podzol, Arenosol and Umbrisol to the lower extent than CS1, and increased the mechanical strength of Fluvisol. Similarly, as for CS1, the intensity of these changes depended on chitosan dose and on the number of wetting–drying cycles with which the soil–chitosan mixtures were pretreated. Exemplary stress–strain curves of soil cylindrical aggregates containing CS2 are shown in Figure 2.

The maxima of the dependencies shown in Figure 1 and Figure 2, expressing a load per unit average of the cross-sectional area at which the cylindrical specimen of soil fails under compression during the Unconfined Compression Strength, UCS [Pa], are presented in Figure 3.

The UCS of the aggregates containing CS1 generally decreases with the increase in chitosan percentage. A similar effect is observed for CS2 amended Umbrisol. Both acidic soils (Podzol and Arenosol) amended with CS2 behave in a different way: after the initial drop, the UCS increases with increasing chitosan dose, finally approaching the UCS value for the nontreated soil. Quite opposite behavior is noted for CS2 amended Fluvisol, for which the UCS seems to tendentially increase with an increase in the chitosan rate.

The UCS of neutral and alkaline soils (Fluvisol and Umbrisol, respectively) after 10 wetting/drying cycles is lower than the UCS of these soils subjected to a single cycle, whereas the increase in the number of wetting–drying cycles increases the UCS of acidic soils, Podzol and Arenosol, containing higher doses of both kinds of chitosan.

Similar dependencies were observed for Young’s modulus and chitosan percentage, since, as commonly observed, UCS and Young’s modulus are roughly proportional to each other, as it is presented in Figure 4 for the studied soils.

Structural porosity is considered to be the best measure of the susceptibility of soils and other granular materials to mechanical damage: more porous aggregates would break easier. The porosities of the studied soil/chitosan aggregates, preconditioned with one or ten wetting–drying cycles, are presented in Figure 5.

In general, the addition of both types of chitosan causes a loosening of the soil structure (increasing porosity) with an exception of Fluvisol amended with CS2, for which chitosan addition decreased soil porosity and made the soil more compact. It was observed that CS1 increased soil porosity more than CS2 and the structure became less porous with an increase in the number of wetting–drying cycles.

## 3. Discussion

Changes in UCS due to the addition of CS1 accompany the opposite changes in porosity for all soils. This trend is valid also for CS2 amended Fluvisol and Umbrisol. However, for very acidic Podzol and acidic Arenosol despite the porosity tendentially increasing with increasing chitosan dose, after the initial drop, the UCS increases also. The increase in porosity observed in the vast majority of the studied soil/chitosan aggregates is most probably caused by the loosening of the soil structure by coarser chitosan particles. However, Kubavat et al. [49] mentioned about soil porosity increase also after the addition of nanoparticles of chitosan copolymerized with methacrylic acid. The higher effect of CS1 on the porosity increase may indicate that CS1 is composed of larger particles than CS2. The exceptional decrease in porosity and a simultaneous increase in UCS for Fluvisol amended with CS2 can be probably connected with the highest content of clay in this soil. In the control soil, the total volume of porous conglomerates of clay particles may exceed the total volume of the soil skeletal pores (free spaces between coarser soil particles) and the coarser particles are “suspended” in the clay phase. In such a case, the addition of nonporous chitosan particles replaces porous clay conglomerates and so the volumetric porosity decreases. In the “suspended” state, rigid skeletal particles rarely contact each other and the mechanical stability of such a system approaches that of the pure clay itself. Additional skeletal porosity created by coarser chitosan particles may be consecutively filled by clay and more contacts between coarser particles occur, and so, the mechanical strength increases. As reported by Jozefaciuk et al. [50] and Horabik and Jozefaciuk [51] there exists a maximum of UCS at a point where all free spaces between coarser particles are totally filled by clay conglomerates and no “free” clay excess is present. Such be the case, increasing the volume of clay makes the UCS smaller. The fact that a similar phenomenon does not occur for CS1 may be because CS1 particles are larger than those of CS2, and, only with the smallest CS1 additions does the volume of newly created free skeletal spaces exceed the volume of the excess of the clay which is available for their total filling.

The other exceptional behavior which needs more detailed examination is that in soils of low pH (Podzol and Arenosol) higher CS2 doses induced the increase in UCS. Electrostatic bonds of positively charged chitosan in a low pH range (PZC of chitosan is usually reported to occur at pH values around 6.5–7.6 [52,53,54]) with negatively charged soil particles may be responsible for this effect. In soils of higher pH, chitosan around its PZC develops either a small amount of positive charge or a small amount of negative charge on a small surface area, and the electrostatic forces that are generated with negatively charged soil components can range from weakly attractive to weakly repulsive. Additionally, different dissolution/jellification patterns of both studied types of chitosan may exert a significant effect on their interactions with soil components because chitosan gel of highly extended surface can form much more electrostatic bonds than chitosan particles of much smaller surface area. Because CS2 of lower molecular mass dissolves better and faster in soil organic acids (fulvic and humic acids) than CS1 having a higher molecular mass, the effect on UCS increase was observed only in the case of CS2. In a jellified form, chitosan can also glue soil particles together by adhesive forces or by the formation of polymer ties connecting neighboring soil particles which are not in direct contact [55]. Differences between the action of both kinds of chitosan on soils are apparently connected to their molecular properties. As it has been frequently reported, molecular weight and molecule size of polymers play a key role in defining the nature of interactions with various soils and soil minerals. Moen and Richardson [56] found that small-sized polymers distribute more evenly in the microaggregate fraction of soils because of their greater ability to penetrate the fine pores. Richardson et al. [57] also found that high molecular weight polymers may maximize soil–polymer interactions; however, the effectiveness could be affected by limited polymer penetration of the soil surface and failure to attain uniform aggregation adsorption. On the other hand, they found that small-sized polymers could create a more homogeneous soil stabilizing polymer network.

The effect of wetting–drying cycles on soil mechanical resistance depended both on the soil reaction and the kind of chitosan. Ambient solution pH and concentration affect the surface charge and conformation of charged polymers and therefore influence polymer adsorption onto soil particles. A higher concentration of the polymer solutions enables sufficient active functional and structural groups in the polymer to be available for interaction with soil particles and therefore could increase the efficiency of polymer stabilization [45]. Since solubilization and gelling are time-dependent, an increase in the mechanical stability of soil–chitosan mixtures in time occurs in acidic soils. In contrast, for neutral Fluvisol and alkaline Umbrisol, wherein chitosan around its PZC is hardly soluble [58], increasing the number of wetting/drying cycles leads to a decrease in UCS. A similar decrease in the strength of biopolymer treated soils with increasing wetting–drying cycles has been frequently reported [48,59].

As was observed by Adamczuk et al. [60] the water stability of aggregates containing CS1 preconditioned with a single cycle of wetting–drying was generally lower than for aggregates of the control soils. After ten wetting–drying cycles, these aggregates became more water-resistant. In contrast to CS1, the water stability of CS2-containing aggregates reached high water resistance just after the first drying–wetting cycle and it increased only slightly after the next nine wetting–drying cycles. Both types of chitosan increased soil water repellency, which increased further after 10 wetting–drying cycles. The wettability for CS1 amended soils was higher than for CS2. Both water stability and wettability may increase because of the high wetting angles of chitosan, according to Cassie’s law [61]. The effect of chitosan on increasing water stability and water repellency might have been caused also by adsorption of the dissolved chitosan molecules on soil components. Even very small amounts of large chitosan molecules can cover the surface of soil particles to a great extent, forming hydrophobic patches. It is possible that this mechanism may decrease the mechanical stability of the aggregates, as well. If, as is likely, the surface covered by chitosan molecules becomes flatter (the surface roughnesses are levelled), the internal friction between smoother soil particles is reduced, and, in consequence, the UCS becomes smaller. It is probable that such an occurrence may be proven by studies of the fractal dimension of chitosan adsorbed soil (a decrease in fractal dimension would indicate surface flattening), which will be a problem in further studies.

Summarizing: coexistence of several antagonistic and synergic mechanisms can explain the observed impact of chitosan on the mechanical and water stability, and wettability of soils. The first is the loosening of soil structure (increase in porosity) due to the addition of coarse chitosan particles that can decrease both mechanical and water stability. However, in clayey soils addition of coarser chitosan particles can make the soil more mechanically resistant. The second mechanism is the formation of electrostatic bonds between positively charged chitosan particles (or jellified/dissolved chitosan molecules) and negatively charged soil components leading to an increase of both mechanical and water stability. This phenomenon would increase with a decrease in soil pH due to better charging and solubilization of chitosan at low pHs. The third mechanism is the formation of adhesive bonds between chitosan and soil components, which, similarly to the previous mechanism, should depend on the amount of the dissolved/jellified chitosan. The fourth mechanism is the adsorption of chitosan molecules on surfaces of soil components causing an increase in soil water stability and hydrophobicity of the soil material along with a possible reduction of the mechanical stability of the soil due to decreasing surface roughness and internal friction. The intensity of all aforementioned mechanisms depending on chitosan jellification, solubilization, surface charge and chain stiffness are governed by the ambient pH [44] and on physicochemical properties of a given kind of chitosan.

## 4. Materials and Methods

Two kinds of chitosan selected to differ in molecular mass and deacetylation degree were used. The first, abbreviated as CS1, was provided by Sigma Aldrich (St. Louis, MO, USA) and the second (CS2) was provided by Beijing Be-Better Technology Co., Ltd. (Beijing, China). Both forms of chitosan were composed of equal parts of the 0.105–0.053 and 0.177–0.105 mm dry sieved fractions. Soil samples were taken from upper 5–15 cm layers of four soils localized in East Poland, air-dried and screened by a 1 mm sieve. The basic properties of the studied soils and types of chitosan are presented in Table 1. This table recalls data reported by Adamczuk et al. [60] who studied water stability and wettability of the same materials as used in the present paper.

### 4.1. Preparation of Soil/Chitosan Aggregates

The soil samples were air-dried and screened by a 1 mm sieve (mesh 18). Carefully homogenized water-saturated pastes were prepared from mixtures of the soils and the chitosan. Distilled water was used for pastes preparation. The content of CS1 and/or CS2 in the mixtures was 0 (control), 0.5, 1, 2, 4, and 8%. Cylindrical aggregates of 20 mm height and 10 mm diameter were formed from the pastes using plastic forms. The first set of aggregates was prepared just after the preparation of the paste and then air-dried (one cycle of wetting–drying) and the second set from the pastes was subjected to 10 wetting–drying cycles (6 days per cycle). All aggregates were then dried until constant mass at laboratory conditions (relative humidity around 60% and temperature around 25 °C). The aggregates are abbreviated further using the abbreviation of a given soil (see Table 1) followed by the number of wetting/drying cycles (e.g., POD1 and POD10 denote aggregates formed from Podzol preconditioned with one and ten wetting/drying cycles, respectively).

### 4.2. Studies of Soil/Chitosan Aggregates

Unconfined compression tests were performed for ten replicates of each aggregate using the Lloyd LRX material testing machine (Bognor Regis, UK). An aggregate placed vertically on the machine basement was pressed by a piston. The load measured with the accuracy of 0.05 N against displacement of the piston moved with the lowest apparatus speed of 10^−5^ m·s^−1^ was registered. The dependence of the compression stress [Pa], (load divided by the aggregate cross-sectional area) versus strain, ΔL/L (relative aggregate deformation, equal to piston displacement divided by the aggregate height) was calculated. From these dependencies values of the Unconfined Compression Strength UCS [Pa] (maximum load per unit average cross-sectional area at which the cylindrical specimen of soil falls in compression) and Young’s modulus, E [Pa] (slope of the linear parts of the stress–strain dependencies) were estimated.

Volumetric porosity (volumetric fraction of pores on the total volume), P, [cm^3^ cm^−3^] of the aggregates were estimated as follows: at first, the volumes of the solids (soil + chitosan) present in 1 cm^3^ of the aggregates, V_s_ [cm^3^], were calculated based on the chitosan percentage in the soil–chitosan mixtures (CS%), the respective particle densities (PD_soil_ and PD_CS1_ or PD_CS2_, presented in Table 1) and the aggregate bulk densities, BD [g cm^−3^] reported in [60]:V_s_ = 1 − [BD∙(1 − CS%/100)/PD_soil_ + BD∙(CS%/100)/PD_CS_] (1)
and next the porosity was calculated as:P = 1 − V_s_. (2)

### 4.3. Statistical Analysis

Average values ± standard deviations were calculated.

## Figures and Tables

**Figure 1 molecules-27-02273-f001:**
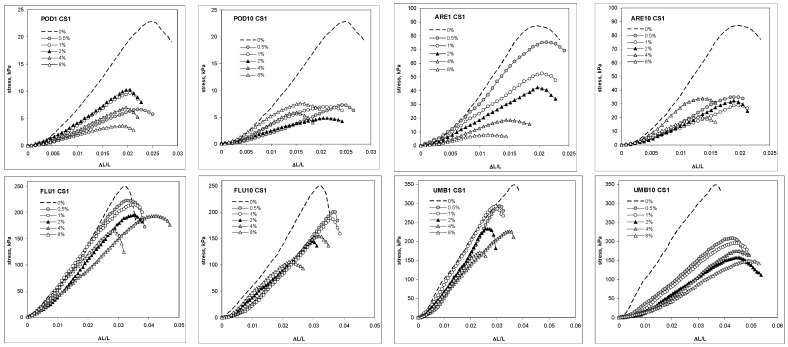
Exemplary compression stress-dilatation dependence for the aggregates of Podzol (POD), Fluvisol (FLU), Umbrisol (UMB) and Arenosol (ARE) amended with different amounts of chitosan CS1, preconditioned with 1 or 10 cycles of wetting–drying. The number of the cycles is written after the abbreviation of each soil.

**Figure 2 molecules-27-02273-f002:**
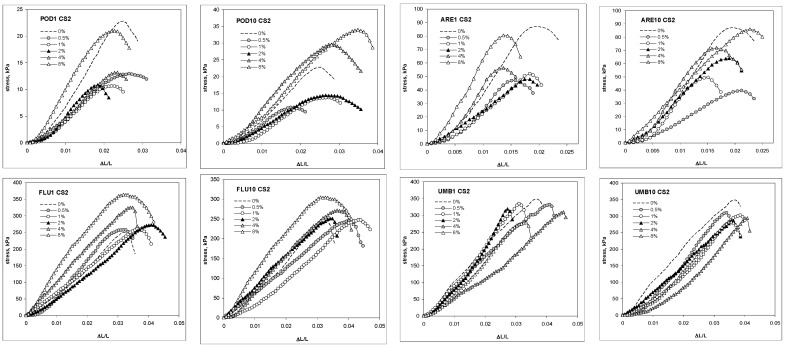
Exemplary compression stress-dilatation dependence for the aggregates of Podzol (POD), Fluvisol (FLU), Umbrisol (UMB) and Arenosol (ARE) amended with different amounts of chitosan CS2, preconditioned with 1 or 10 cycles of wetting–drying. The number of the cycles is written after the abbreviation of each soil.

**Figure 3 molecules-27-02273-f003:**
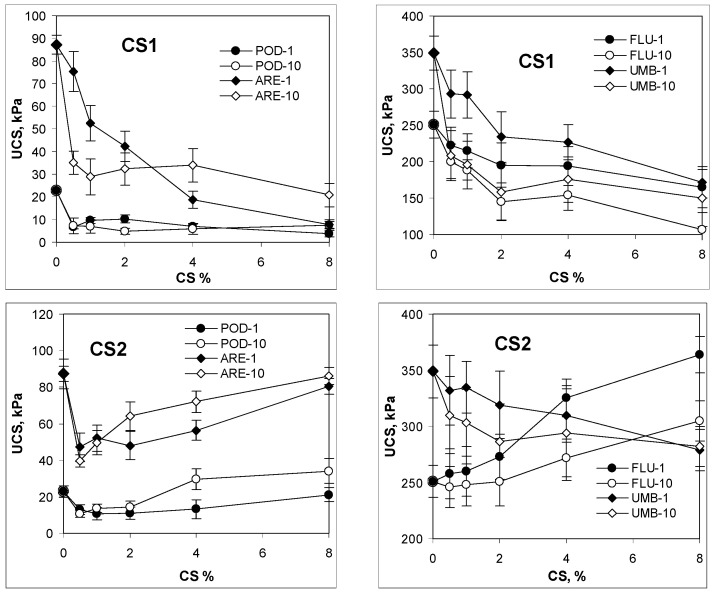
Average values of unconfined compressive strength of the studied aggregates of Podzol (POD), Fluvisol (FLU), Umbrisol (UMB) and Arenosol (ARE) amended with different amounts of chitosan (CS1 and CS2), preconditioned with one or ten cycles of wetting–drying. The number of the cycles is written after the abbreviation of soils. Error bars depict standard deviations.

**Figure 4 molecules-27-02273-f004:**
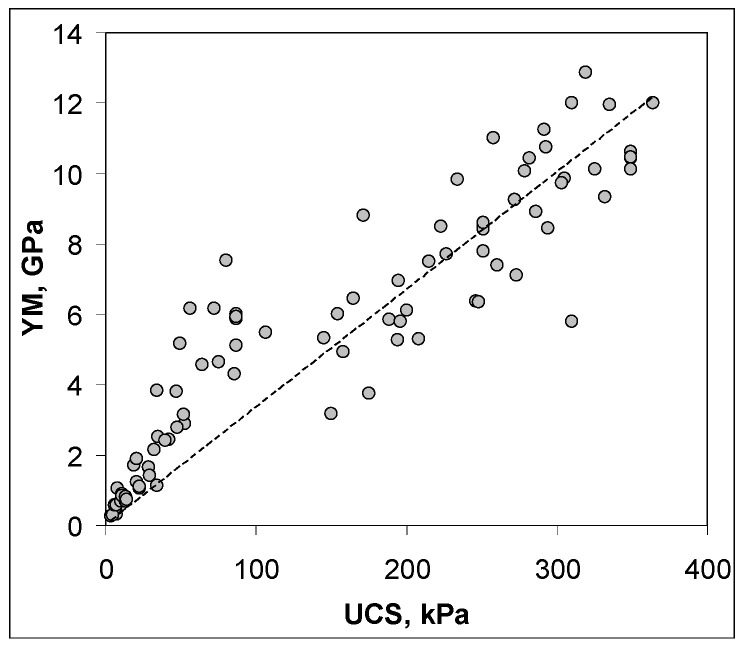
Dependence of Young’s (YM) modulus and unconfined compressive strength (UCS) for all studied aggregates.

**Figure 5 molecules-27-02273-f005:**
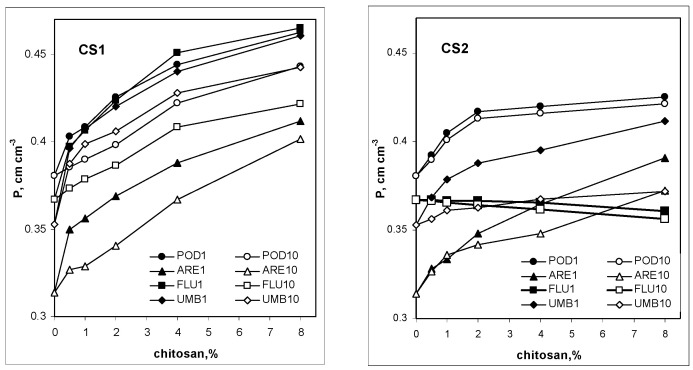
Dependencies of the porosity (P) of soil aggregates on the amendment of the studied chitosan (CS1 and CS2). The soils are abbreviated as: POD—Podzol, ARE—Arenosol, FLU—Fluvisol, UMB—Umbrisol. The number after the soil abbreviation shows the number of wetting–drying cycles applied to soil aggregates. Error bars are covered by the labels of the points.

**Table 1 molecules-27-02273-t001:** Characteristics of the studied soils and chitosan.

Abbreviation	POD	ARE	FLU	UMB	CS1	CS2
Material	Podzol	Arenosol	Fluvisol	Umbrisol	Chitosan1	Chitosan2
Locality E	22°58′41″	22°26′6″	22°59′38″	21°41′59″		
Locality N	51°9′14″	51°2′9″	51°9′43″	50°49′25″		
pH	4.1	5.5	6.5	7.7		
Particle density [g cm^−3^]	2.54	2.62	2.62	2.68	1.51	1.54
Nitrogen [%]	0.16	0.13	0.46	0.14	7.51	7.79
Total organic carbon [%]	0.65	1.55	3.04	0.9	41.59	41.27
Sand (0.063–2 mm) [%]	72.4	47.1	20.2	10.4		
Silt (0.002–0.063 mm) [%]	25.9	46.2	52.2	72.4		
Clay (<0.002 mm) [%]	1.7	6.7	27.6	17.2		
Grain fraction 0.177–0.105 mm [%]					50	50
Grain fraction 0.105–0.053 mm [%]					50	50
Degree of Deacetylation (DD)					0.77	0.91
Average molecular mass (M), [kDa]					699	280

## Data Availability

All data are available from authors after a reasonable request.

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
