# Peer review of "Impact of Chitosan on the Mechanical Stability of Soils"

_molecules, 2022, doi:10.3390/molecules27072273_

Round 1

Reviewer 1 Report

The article is interesting. However, it is necessary that the authors explain in detail why the two samples of chitosan to be used in the experiments were chosen.

Author Response

Dear Reviewer 1

Thank you very much for your opinion and comments. Our corrections are marked red in the revised MS.

The article is interesting. However, it is necessary that the authors explain in detail why the two samples of chitosan to be used in the experiments were chosen

We have a small collection of chitosans of different properties. We chose two chitosans which had the smallest and the highest molecular mass among them.

Reviewer 2 Report

Interesting work on the topic Impact of chitosan on the mechanical stability of soils. The manuscript is within the scope of the Journal but needs major revision.

  • I cannot find any novelty in this work. This is the main issue related to this script.
  • The abstract section needs to be clearly provided. Its looks very confusing. Kindly rewrite this section again.
  • Under the Introduction section, the literature review is very poorly covered.
  • The author should clearly describe the one or more key hypotheses that the work described in the manuscript was intended to confirm or refute. The inclusion of a hypothesis statement makes it simple to contrast the hypothesis with the most relevant previous literature and point out what the authors feel is distinct about the current hypothesis (novelty).
  • The main reason for choosing chitosan needs deeper discussion and proper justification.
  • Materials details need to be added.
  • Provide molecular weight of chitosan used in the present work. How the different molecular weights of chitosan impact the application.
  • The results and discussion section are too weak. Need more detailed in-depth discussion.

Author Response

Dear Reviewer 2

Thank you very much for your opinion and comments. Our corrections are marked red in the revised MS.

  • Interesting work on the topic Impact of chitosan on the mechanical stability of soils. The manuscript is within the scope of the Journal but needs major revision.
  • I cannot find any novelty in this work. This is the main issue related to this script.

The novelty in this work is that for soil mechanical resistance studies we used chitosan in powder form what was studied for the first time. Frankly speaking we found only a few papers on mechanical stability of soils amended with pure chitosan (in jellified form). This is thoroughly explained in the new Introduction section, rewritten completely.

  • The abstract section needs to be clearly provided. Its looks very confusing. Kindly rewrite this section again.

The abstract was rewritten slightly, however, since the Editorial demand is to write Abstract using no more than 200 words we tried to include in it all necessary information, including research area, novelty of the approach, materials, methods and results. The main results, shorten, however, to a large extent, were included in the second half of the Abstract. It is not possible to extend the latter fragment not exceeding 200 words.

  • Under the Introduction section, the literature review is very poorly covered.

We wrote the new Introduction to fulfil this and the previous demand.

  • The author should clearly describe the one or more key hypotheses that the work described in the manuscript was intended to confirm or refute. The inclusion of a hypothesis statement makes it simple to contrast the hypothesis with the most relevant previous literature and point out what the authors feel is distinct about the current hypothesis (novelty).

Our main hypothesis was that chitosan added in the powder form should affect soil mechanical resistance and that this should depend on soil pH and deacetylation degree and molecular mass of the chitosan. However, any positive or negative result could confirm this hypothesis since it is to broad to be proved or denied. This is because almost nothing has been reported in this area, probably because all researches know that powdered chitosan does not consolidate soils, and most of the studies were performed for soil consolidation. Instead to place the above hypothesis, we simply wrote that there exists a knowledge gap that is tried to be filled by our work.

  • The main reason for choosing chitosan needs deeper discussion and proper justification.

We chose chitosan because it is increasingly introduced to soils, mainly in its powder form and, since its effects on soil mechanical properties have been not studied up to date, we thought that it is interesting to know what is going on.

  • Materials details need to be added.

We added some more information to M&M section.

  • Provide molecular weight of chitosan used in the present work. How the different molecular weights of chitosan impact the application.

Chitosan molecular weights are presented in Table 1, M&M  section.

  • The results and discussion section are too weak. Need more detailed in-depth discussion.

We extended the Discussion section including more details supported by new citations.

Reviewer 3 Report

Adamczuk and G. Józefaciuk study the impact of chitosan on compression resistance of different soil (Podzon, Arenosol, Fluvisol and Umbrisol). Factors such as chitosan concentration and type or number of wetting-drying cycle were considered in the evaluation of mechanical properties of treated soils.

The article is the continuation of the work “Adamczuk, A.; Kercheva, M.; Hristova, M.; Jozefaciuk, G. Impact of Chitosan on Water Stability and Wettability of Soils. Materials 2021, 14, 7724. https://doi.org/10.3390/ma14247724” where the same materials were used to study the water stability and wettability of soils, while in this manuscript the mechanical resistance was investigated.

In my opinion, it would have been more correct to submit the manuscript to the same journal.

Moreover, half of the results and discussions derives from data reported in the previous article. Since there is a close interconnection between this article and the previous one it would be better to merge these mechanical data together with those of wettability and water stability in one article.

Despite this, I consider the article worthy of publication because gave new information about the use of biomaterials in the soil treatment. I suggest its publication after some minor revision

Author Response

Dear Reviewer 3

Thank you very much for your opinion and comments. Please find below our replies. Our corrections are marked red in the revised MS.

Adamczuk and G. Józefaciuk study the impact of chitosan on compression resistance of different soil (Podzon, Arenosol, Fluvisol and Umbrisol). Factors such as chitosan concentration and type or number of wetting-drying cycle were considered in the evaluation of mechanical properties of treated soils.

The article is the continuation of the work “Adamczuk, A.; Kercheva, M.; Hristova, M.; Jozefaciuk, G. Impact of Chitosan on Water Stability and Wettability of Soils. Materials 2021, 14, 7724. https://doi.org/10.3390/ma14247724” where the same materials were used to study the water stability and wettability of soils, while in this manuscript the mechanical resistance was investigated.

In my opinion, it would have been more correct to submit the manuscript to the same journal.

We thought to submit the paper to Materials, however we decided to try a submission to Molecules because the latter journal edits a special issue devoted to chitosan where our manuscript fits exactly.

Moreover, half of the results and discussions derives from data reported in the previous article. Since there is a close interconnection between this article and the previous one it would be better to merge these mechanical data together with those of wettability and water stability in one article.

You are right that larger article could be more interesting. However, we decided to submit the article on water stability as early as possible, to be among the first who reported such studies (probably it is a common dream of all researchers to be in a such group). At that moment we just started mechanical stability experiments.

Despite this, I consider the article worthy of publication because gave new information about the use of biomaterials in the soil treatment. I suggest its publication after some minor revision:

- Please avoid informal and personal writing forms such as personal pronouns “we” “our” and using judgemental words that indicate your feeling about a subject such as “it was interesting for us”(line 49) “it is worth noting” line 100 and 203 . Moreover, avoid the use of term such as “above” to indicate a figure/table that is above the text; instead write the number of figure/table which you refer to.  I also suggest an English language revision by a mother tongue.

We rewrote the text accordingly, however we could not find any native English speaker who deals with scientific problems.

- Materials and method: Please report from the previous article the source of all the used materials Chitosans: “Two different kinds of chitosan were used. The first, abbreviated as CS1, was provided by Sigma Aldrich (St. Louis, MO, USA) and the second (CS2) was provided by Beijing Be-Better Technology Co., Ltd. (Beijing, China)” Soils: “Soil samples were taken from upper 5-15 cm layers of four soils localized in East Poland, air dried and screened by a 1 mm sieve”. I also suggest to report in the table the locality and/or the longitude of the soils.

We added the requested data.

Line 227. This test follows some normative? If not, can you justify the choice of the speed of 10-5 m/s? It simulates some natural phenomenon that the soil undergoes?

Generally at lower piston speeds, better results are obtained. This was the lowest speed available for our apparatus. It is mentioned in the text.

Line 235 Volumetric porosity is reported as [m3 m-3 ], in Figure 5 is [cm cm-3 ] while in equation (2) P=1-Vs and Vs in line 235 is defined as the “volumes of the solids in 1 cm3 of aggregates”, so I suppose [cm3 /cm3 ] that brings to a non-dimensional value (it is a volumetric fraction of pores on the total volume). Please homogenize different measure units/definitions.

The units are homogenized.

Line 117 I think that 4 experiments are not a rule but a scientific evidence.

We corrected the respective statement.

It was demonstrated that in different type of soil (from acidic to alkaline) act as a “porosity reducer”. As you report CS2 probably works thanks to its smaller dimension. In very fine grounded soils maybe also the CS2 will have issues to reduce the porosity of the soils.

We like to check it experimentally using more samples of  chitosans and soils.

Reviewer 4 Report

The manuscript deals with the impact of chitosan on mechanical stability of soils.

The manuscript is interesting. Nevertheless, the authors have the following published article “Adamczuk, A.; Kercheva, M.; Hristova, M.; Jozefaciuk, G. Impact of Chitosan on Water Stability and Wettability of Soils. Materials 2021, 14, 7724. https://doi.org/10.3390/ma14247724”. The authors must revise the manuscript to avoid any similarities.

The English language must be revised.

Abstract

This section must be improved. Please present your main results.

Introduction

The topics must be better linked.

Materials and methods

A statistical analysis section is missing.

Author Response

Dear Reviewer 4

Thank you very much for your opinion and comments. Please find below our replies. Our corrections are marked red in the revised MS.

The manuscript deals with the impact of chitosan on mechanical stability of soils.

The manuscript is interesting. Nevertheless, the authors have the following published article “Adamczuk, A.; Kercheva, M.; Hristova, M.; Jozefaciuk, G. Impact of Chitosan on Water Stability and Wettability of Soils. Materials 2021, 14, 7724. https://doi.org/10.3390/ma14247724”. The authors must revise the manuscript to avoid any similarities.

The avoidance of similarities between both papers is not possible, because the materials used are the same, as it was explicitly stated in the present submission. Therefore, to get Readers informed about the materials properties it was necessary to describe them. Moreover, wanting to have a broader view on chitosan action on soils, we discussed also connections between water and mechanical stability. To do this, some similarities were necessary also.

 The English language must be revised.

We tried to do our best to improve English.

Abstract

This section must be improved. Please present your main results.

Since the Editorial demand is to write Abstract using no more than 200 words we tried to include in it all necessary information, including research area, novelty of the approach, materials, methods and results. The main results, shorten, however, to a large extent, were included in the second half of the Abstract. It is not possible to extend the latter fragment not exceeding 200 words.

Introduction

The topics must be better linked.

We completely rewrote The Introduction to fulfil this request.

Materials and methods

A statistical analysis section is missing.

We used only a simple statistics, described in the recent MS.

Round 2

Reviewer 2 Report

The authors have satisfactorily revised the manuscript.